# Semi-supervised audio tagging with deep co-training and augmentations

## Abstract

In this work, we explored the task of audio tagging in a semi-supervised context. The recently proposed Deep Co-Training (DCT) algorithm has shown impressive results in visual object recognition and outperformed other semi-supervised state-of-the-art methods such as Mean Teacher and GANs. DCT uses two or more deep neural networks and adversarial examples to enforce complementarity between the models trained on the same data. We adapted DCT to audio tagging, and we report experiments on the publicly available UrbanSound8K dataset. We compare models trained with 10% of labeled data using supervised training and using DCT, which may benefit from the remaining 90% unlabeled data. To go further than the original DCT proposal, we propose to artificially increase the 10% of labeled files by simply duplicating them in the mini-batches during learning, and transforming them with audio data augmentations. If standard DCT already showed performance gains against supervised learning (17% relative gain), the use of duplication combined with data augmentations on the labeled examples lead to additional significant performance improvements (26% gain)[1].

## 1. Introduction

The availability of large datasets of audio data, such as AudioSet (Gemmeke et al., 2017), allows the creation of large deep neural networks with more generalization capability. Yet, collecting this data is more costly, both financially and in terms of time. Automatic tools which are based on public and open annotations bring noise in the labels and reduce the overall label quality.

---

[1]Anonymous Institution, Anonymous City, Anonymous Region, Anonymous Country. Correspondence to: Anonymous Author <anon.email@domain.com>.

Preliminary work. Under review by the International Conference on Machine Learning (ICML). Do not distribute.

[1]Source code is available at https://github.com/leocances/Deep-Co-Training.git

Semi-supervised learning (SSL) aims to reduce the need for labeled data. The unlabeled data are used during training to guide the model to a better generalization on unseen data. This is specifically important regarding annotated data availability and variability for most audio event types. Semi-supervised learning also reduces the need on annotated samples in a dataset, reducing its cost and creation time. This approach is widely observed in computer vision tasks, and started few years ago in audio domain.

In this work, we focus on audio tagging (AT) in a semi-supervised setting. AT consists of automatically identifying sound events in recordings by inferring global labels called tags. It is often an essential subtask of Sound Event Detection (SED) application.

The goal is to reach the same performance of a model trained on the full set of labeled data in a supervised fashion, by using only parts of the data as labelled. More specifically, we explore the use of DCT to perform AT with Convolutional Neural Networks (CNN). DCT, an extension to deep learning of the highly acclaimed Co-Training generic framework for semi-supervised learning (Blum & Mitchell, 1998), has been recently proposed by Qiao and colleagues (Qiao et al., 2018). The authors obtained impressive results in visual object recognition and showed that DCT outperforms other deep learning competitive approaches, such as Mean Teacher (Tarvainen & Valpola, 2017).

We test DCT on UrbanSound8K (Salamon et al., 2014), a publicly available dataset well-suited for AT. Since this dataset is comprised of labeled data only, we simulate unlabeled data by using only 10% of the training subset as labeled data and the remaining 90% as unlabeled data. Doing so allows us to monitor how well DCT is behaving during training and the performance of our system on the unlabeled subset.

## 2. Related work in SSL for audio tagging

Semi-supervised learning (SSL) aims at improving classification accuracy by using unlabeled data in addition to labeled data. It is sometime coupled with self-supervised learning like ReMixMatch (Gidaris et al., 2018)

Some SSL approaches for image recognition were adapted to audio-related tasks such as pseudo labeling (Lee, 2013),

mean teacher (Tarvainen & Valpola, 2017), or more recently guided learning (Lin & Wang, 2019).

Nguyen and colleagues (Nguyen et al., 2018) propose to use pseudo-labeling for automatic label verification on the unsupervised part of their subset. They use label smoothing to further reduce over-fitting. In (Dorfer & Widmer, 2018), pseudo-labeling is also used but in an iterative process to annotate unsupervised files that have been classified with high confidence. More concretely, they used pseudo-labeling to verify possibly noisy labels, by comparing the labels of unverified examples with the predictions of a neural network, which can be interpreted as a version of "supervised" pseudo-labeling. However, it is an iterative process and can introduce incorrect annotation if the model misclassifies the unlabeled samples.

We can find applications of student-teacher and mean teacher (Tarvainen & Valpola, 2017) in the DCASE 2018 and 2019 task 4 challenge on weakly-supervised SED. The 2018 winners trained CNNs on both a small labeled subset and a larger unlabeled one (JiaKai, 2018). They used a Student-Teacher approach, in which two models are built in a way that makes them complementary and more robust.

More recently, ReMixMatch (Berthelot et al., 2019) applies a random rotation on strongly augmented images. The model should then be able to predict which rotation angle is applied to the input image. This self-supervised loss is then added to the semi-supervised loss.

To the best of our knowledge, the work reported in this article is the first use of DCT to perform semi-supervised audio tagging.

## 3. Deep co-training (DCT) overview

DCT has been recently proposed by Qiao and colleagues (Qiao et al., 2018). It is based on Co-Training (CT), the widely acclaimed generic framework for semi-supervised learning proposed by (Blum & Mitchell, 1998). Co-training's main idea relies on the assumption that each data point has two *views*, and that each view is sufficient to train a separate model, using labeled data. Predictions are made with the two models on the examples of an unlabeled set and the examples with highest confidence are selected and used to augment the training labeled subset, in an iterative process.

DCT is an adaptation of CT in the context of deep learning. Instead of relying on views of the data that are different (ideally, the two views are conditionally independent given the class), DCT makes use of adversarial examples. The unlabeled subset makes up for large part of each mini-batch during the training. Doing so, it avoids the long iterative process.

Let $\mathcal{S}$ and $\mathcal{U}$ be the subsets of labeled and unlabeled data, respectively, and let $f$ and $g$ be the two neural networks that are expected to collaborate.

The DCT loss function is comprised of three terms, as shown in Eq. 1. These terms correspond to loss functions estimated either on $\mathcal{S}, \mathcal{U}$, or both. Note that during training, a mini-batch is comprised of labeled and unlabeled samples in a fixed proportion. Furthermore, in a given mini-batch, the labeled examples given to each of the two models are different.

$$\mathcal{L} = \mathcal{L}_{\text{sup}} + \lambda_{\text{cot}}\mathcal{L}_{\text{cot}} + \lambda_{\text{diff}}\mathcal{L}_{\text{diff}} \quad (1)$$

The first term, $\mathcal{L}_{\text{sup}}$, given in Eq. 2, corresponds to the standard supervised classification loss function for the two models $f$ and $g$, estimated on examples $x_1$ and $x_2$ sampled from $\mathcal{S}$. In our case, we use categorical Cross-Entropy (CE), the standard loss function used in classification tasks with mutually-exclusive classes.

$$\mathcal{L}_{\text{sup}} = \text{CE}(f(x_1), y_1) + \text{CE}(g(x_2), y_2) \quad (2)$$

In SSL and Co-Training, the two classifiers are expected to provide consistent and similar predictions on both the labeled and unlabeled data. To encourage this behavior, the Jensen-Shannon (JS) divergence between the two sets of predictions is minimized on examples $x_u$ sampled from the unlabeled subset $\mathcal{U}$ only. Indeed, there is no need to minimize this divergence also on $\mathcal{S}$ since $\mathcal{L}_{\text{sup}}$ already encourages the two models to have similar predictions on $\mathcal{S}$. Eq. 3 gives the JS analytical expression, with $H$ denoting entropy.

$$\begin{aligned}\mathcal{L}_{\text{cot}} = \ & H\Big(\frac{1}{2}(f(x_u) + g(x_u))\Big) \\ & - \frac{1}{2}\Big(H(f(x_u)) + H(g(x_u))\Big) \quad (3)\end{aligned}$$

For DCT to work, the two models need to be complementary: on a subset different from $S \cup U$, examples that are misclassified by one model should be correctly classified by the other model (Krogel & Scheffer, 2004). In deep learning, this can be achieved by generating adversarial examples with one model and train the other model to be resistant to these adversarial samples. To do so, the $\mathcal{L}_{\text{diff}}$ term (Eq. 4) is the sum of the Cross-Entropy losses between the predictions $f(x_1)$ and $g(x'_1)$, where $x_1$ is sampled from $S \cup U$ and $x'_1$ is the adversarial example generated with model $f$ and $x_1$ taken as input. The second term is the symmetric term for model $g$.

$$\mathcal{L}_{\text{diff}} = \text{CE}(f(x_1), g(x'_1)) + \text{CE}(g(x_2), f(x'_2)) \quad (4)$$

For the adversarial examples generation, we use the Fast Gradient Signed Method (FGSM, (Goodfellow et al., 2015)), as in Qiao's work.

For more in-depth details on the technical aspects of DCT, the reader may refer to (Qiao et al., 2018). We implemented DCT exactly as described in Qiao's article, using PyTorch, and made sure to accurately reproduce their results on CIFAR-10: about 90% accuracy when using only 10% of the training data as labeled data (5000 images).

## 4. Experimental setup

### 4.1. Audio material

The UrbanSound8K dataset (Salamon et al., 2014) contains 8732 labeled sound excerpts of urban sounds from 10 classes: air conditioner, car horn, children playing, dog bark, drilling, engine idling, gunshot, jackhammer, siren, and street music. Their duration is up to four seconds for each recording, and the corpus is comprised of 8.7 hours in total. A given class can occur several times within a recording, and the sound classes are mutually exclusive: events of a single class occur in a given recording. The task involved, thus, is called monophonic audio tagging.

The dataset comes into a predefined 10-fold split that is recommended by the authors (Salamon et al., 2014), in order to get comparable results with other solutions. Thus, all the results presented here-after were obtained using 10-fold cross-validation on these splits.

As DCT is a semi-supervised learning method, we artificially split the training subsets into two parts: one labeled part denoted by $\mathcal{S}$ (for supervised) and one unlabeled part denoted by $\mathcal{U}$ (for unsupervised). We nevertheless use the ground truth of the latter to verify our results, but we do not use it during training. The amount of labeled files used for training represents 10% (873 files) of the complete training set.

As input to the networks, 64 log-Mel filter-bank (F-BANK) coefficients were extracted every 25 ms on 50 ms duration frames, with 20 Hz and 11025 Hz as minimum and maximum frequency values to compute the Mel bands, respectively. Hence, for each 4-seconds file, a $64 \times 173$ matrix is extracted. For file smaller than 4 seconds, we apply zero padding at the end of the recordings.

We report performance using standard accuracy averaged on the 10-folds and standard deviation.

### 4.2. Model description

Our model is based on a similar architecture than the one proposed in (Salamon et al., 2014). The model is small, with 62.5 thousand trainable parameters, and allows experiments

to be performed quickly while achieving state of the art performance. Thus, we used it to perform all the experiments described in Section 6. Its architecture is the following:

- L1: 24 filters with a receptive field of (3,3) and (1, 1) padding, followed by (4,2) strided max-pooling and a rectified linear unit (ReLU).

- L2-3: twice 48 filters with a receptive field of (3,3), followed by (4,2)-strided max-pooling and ReLU.

- L4: 48 filters with a receptive field of (3, 3), ReLU (no pooling).

- L5: 10 output units, with a softmax activation function.

Dropout (Srivastava et al., 2014) is applied to the input of the last layer, with probability 0.5.

### 4.3. Training

DCT loss, describe in Equation 1, introduces some hyper-parameters that must be finely tuned to obtain good performance.

For our system, $\lambda_{cot}$, $\lambda_{diff}$, and epsilon $\epsilon$ are respectively equal to 5, 0.25, and 0.1. These $\lambda$ factors are applied to their respective part of the loss and $\epsilon$ is used for the adversarial generation. We train our system using stochastic gradient descent (SGD) (Bottou, 2010) with momentum 0.9 and weight decay 0.001 during 400 epochs and a batch size of 100 samples. We use a cosine learning rate schedule define by $\text{lr} = 0.01 \times (1.0 + cos((T-1) \times \pi/400))$ $\lambda_{cot}$ and $\lambda_{diff}$ follow a cosine warmup on 160 epochs.

## 5. Results

| | Accuracy |
|---|---|
| Supervised | $47.3 \pm 4.1$ |
| Deep Co-Training | $55.4 \pm 4.6$ |
| **Augmented Deep Co-Training** | $\mathbf{59.7 \pm 5.1}$ |

*Table 1.* Categorical accuracy and standard deviation report on the UrbanSound8k predefined 10 folds cross-validation while using 10% of the dataset as labeled. Deep Co-Training brings a gain of 8.1 points and our best system, "augmented" Deep Co-Training, an increase of 12.4 points.

The best system, Augmented Deep Co-Training, has been trained using only 10% of the ground truth available, while the 90% rest was considered unknown. However, each training minibatch was composed of 40% of supervised files. This ratio is reached by duplicating the supervised files. Therefore, the total number of different annotated files does not change.

Pitch shift (see 6.2) augmentation was applied, with 75% chance, to those duplicated files to avoid overfitting. The unsupervised files were left untouched.

## 6. Experiment

If DCT applied to audio tagging already shows an improvement in performance as shown in Table 1, it falls short of what can be observed when applied to the visual object recognition task. To further improve the gain already provided by the DCT, we have carried out a series of experiments.

We will start by analyzing the effect of the number of labeled files present in each minibatch, then observe the influence of specific augmentations according to their chance of being applied. Finally, we will combine the two to maximize their respective impacts.

Since DCT can be rather long, the following experiments are carried on a balanced subsample (10%) of the Urban-Sound8k Dataset. Only the best experimental result will be apply on the full dataset for validation.

### 6.1. Mini-batch supervised ratio

The learning of unlabeled files is possible thanks to the presence of a minimum number of labeled files. The more this number increases, the more the system is able to classify unknown files correctly. Figure 1 shows the result of this experiment with labeled file ratios per minibatch of 10, 15, 20, 30, 40, 50, and 75%.

The performance improvement is significant and reaches a plateau at about 45% accuracy (see Figure 1). A model trained with 50% of supervised file per minibatch is up to 6.7 points more efficient than when the distribution of labeled and unlabeled samples per minibatch is different than the default ratio of 10%. On the other hand, the supervised files are duplicated five times, and over-fitting is inevitable.

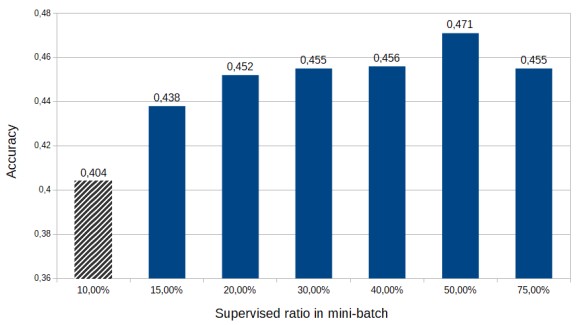

*Figure 1.* Evolution of the accuracy as the ratio of labeled files per minibatch increases. Experiment realized on the sub-sampled dataset (10%)

### 6.2. Augmentation of the supervised subset

To overcome the problem of over-fitting, we augment the annotated files with different signal processing algorithms, such as pitch shift or noise addition. The results of this combination are shown in Figure 2.

The increase in the number of labeled files by mini-batch and the application of augmentation on these duplicate signals significantly enhances system performance. The best score is observed when 40% of the minibatch is supervised with one chance out of two to apply a pitch shift on the labeled samples.

The different augmentation tested, some taken from (Salamon et al., 2014), are describe bellow:

- Pitch Shifting (PS): raise or lower the pitch of the audio sample, Each sample was pitch shifted by 4 values (in semitones): -3, -2, 2, 3.

- Noise (N): A background noise with a Signal Noise / Ratio (SNR) of 20db.

- SpecAugment Dropout: where, on the one hand, 1 to 2 chunks of size varying between 8 and 11 frames is set to zero, and on the other hand, 1 to 2 chunks of size ranging between 4 and 8 mel-bands is also set to zero.

- SpecAugment Stretch: Where chunk of size varying from 5 to 16 frames and 4 to 8 mel-bands could be stretch/compress. After dividing the sample into chunks, each one had a probability of being stretched of 30% with a factor randomly picked in [0.8, 1.2].

When the labeled files are duplicated four times (40%), but the augmentation has a one in two chance of being applied, then statistically, original data are presented to the system twice, encouraging over-fitting. This phenomenon is exacerbated when the percentage of labeled files in each minibatch increases. A way to mitigate this behavior is to raise the chance of applying the augmentation as the percentage of labeled sample increases in the minibatch. The result of this experiment is presented in Figure 3 and is realized using the full dataset.

## 7. Conclusion

In this article, we reported SSL audio tagging experiments carried out on UrbanSound8K, a publicly available dataset. We adapted the Deep Co-Training framework, initially proposed for visual object recognition, to audio tagging.

If the performance of DCT alone showed a significant performance gain, virtually increasing the supervised subset proportion in minibatches while applying augmentation allowed to reduce over-fitting, and resulted in better scores.

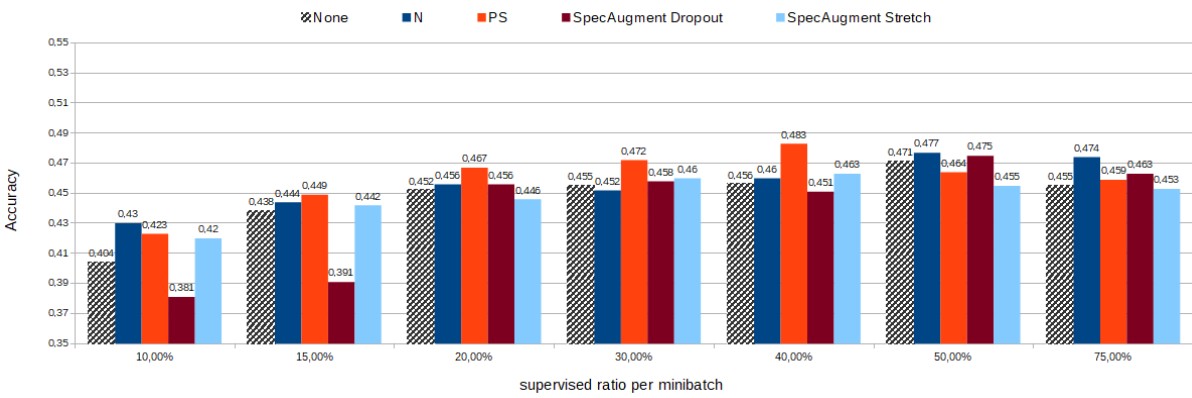

*Figure 2.* Evolution the accuracy when we combine increasing the number of the labeled file in minibatch with some augmentations on these files. The experiment is done on the sub-sampled dataset (10%).

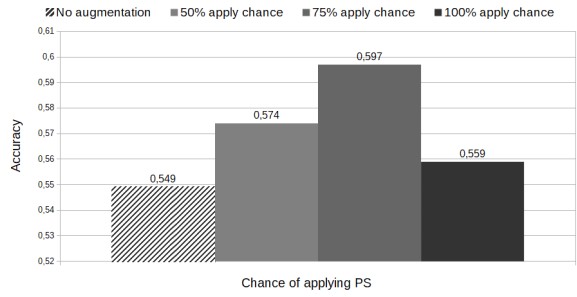

*Figure 3.* Evolution of the accuracy score when increasing the probability of applying the PS augmentation. These results come from the best configuration and calculated on the complete dataset.

Using only 10% of the labeled training files and the remaining data as unlabeled, DCT achieved an accuracy score of 55.4%. When we duplicate some of the labeled files to get a 40% proportion compared to unlabeled samples per minibatch, together with a 75% chance to apply Pitch shift augmentation on these files, the system reached 59.7% accuracy. The difference between supervised learning and "Augmented" DCT, corresponds to 26% relative increase.

There are several lines of work to continue to improve DCT for sound event classification. We plan to confirm the good results obtained with DCT and duplication-augmentation on larger audio datasets, such as DESED (Turpault et al., 2019), for instance. Since DCT takes advantage of multi-view learning, we could use different types of features as input to the network, instead of adversarial examples: the raw signal together with log-Mel features, for example. In Qiao's experiments (Qiao et al., 2018), they show the impact and usefulness of the loss function $\lambda_{\text{diff}}$ as well as the role of adversarial examples. We could perform some tests to validate these observations when using DCT in an audio tagging task. Another recent promising approach regarding efficient audio representations is self-supervised learning approaches, such as PASE+ (Ravanelli et al., 2020). Finally, we plan to compare DCT with other recent SSL algorithms, such as FixMatch (Sohn et al., 2020).

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
