# OpenReview forum: "Semi-supervised audio tagging with deep co-training and augmentations"
_ICML.cc/2020/Workshop/SAS — Submitted to SAS 2020_

### Official Review · AnonReviewer1 · 2020-06-22
**Interesting work but could be explained better**

**Rating:** 4
**Confidence:** 2

**Review:**


Notes:
  -Paper considers semi-supervised audio tagging.

  -Paper uses "deep co-training" which had been successful for visual object recognition.

  -DCT uses two or more deep nets and adversarial examples?

  -Paper uses audio data augmentations (calling it duplication in the abstract is strange, since presumably you'd want to add augmentations as much as possible).

  -Audio tagging = identifying sound events through global labels called tags.

  -Idea: encourage two different views to make similar predictions.

Comments:
  -Mean teacher is actually not that close to the SOTA anymore in SSL.  Interpolation Consistency Training (Verma 2019) is much stronger, and has been improved a few times.  For example, MixMatch (Bertholdt 2019).

  -The paper should mention earlier what the two "views" are for this task.

  -Why do you only make one "pitch augmentation"?  Why not do it on the fly on each iteration, the way normal data augmentation is done?

  -I read the paper and I still don't know what the two views are.

  -Also, in what sense is DCT "adversarial", since it looks like it's matching the predictions of two views.

Review:
From an applied perspective, I felt this paper lacked a sufficiently clear explanation of the different views that would clarify how DCT is working.  Also, I was confused about exactly how data augmentation is applied.

---

### Official Review · AnonReviewer3 · 2020-06-23
**Reviews a cool algorithm, but provides no new theory**

**Rating:** 5
**Confidence:** 4

**Review:**

"Semi-supervised learning also reduces the need on annotated samples in a dataset, reducing its cost and creation time. This approach is widely observed in computer vision tasks, and started few years ago in audio domain." -- The ignorance displayed by this statement is astounding.  The earliest paper on semi-supervised learning in audio, that I know of, is from 1965 (Scudder, Transactions on Information Theory; just a Gaussian signals case, but a very nice proof).  Co-training has been used in audio since the early 2000s; Blum & Mitchell originally proposed it for NLP, so the leap to audio was not too big.  Originally, it was used with different modalities (e.g., audio and vision, or prosody and lexical features of speech).  There were a few papers in 2009 and 2010 that used co-training for audio event detection by forcing the two classifiers to select different subsets of a  common global feature space, then training KNN or other rich classifiers (Yaslan & Catalteppe, 2009, 2010; Zhang, Zhao, Liu and Pang, 2010).  The "Deep Co-Training" algorithm cited in this paper (Qiao et al., 2018)  proposes two pretty cool and important innovations: rather than communicating with one another by passing confidently-labeled examples back and forth, the two views instead communicate by (1)  minimizing Jensen-Shannon divergence between their labelings of unlabeled data, and (2) generating adversarial examples for one another to use in training.

This paper proposes two innovations: (1) use of deep co-training for audio, specifically, for the well-studied UrbanSounds8K corpus, and (2) combination of deep co-training with data augmentation.  These are not huge innovations.  Data augmentation by pitch-shifting and time-stretching was published by two of the original authors of UrbanSounds8K, in their 2017 WASPAA paper (Scaper), where they achieved about 58% F-measure on the UrbanSounds8K corpus.

I am having some difficulty determining whether the reported 59.7% accuracy in this paper should be considred good or bad.  This paper  does  not test data augmentation without deep co-training, so it's not possible to see whether the proposed combination is any better than the use of simple data augmentation by itself.  Fig. 3 shows a downward trend in accuracy for labeled/unlabeled ratios higher than 40 or 50 percent, but the case of 100% is not tested.  (Lu et al., Interspeech 2019) reported 93.4% accuracy on UrbanSounds8K using all labels, or 75.17% using semi-supervised learning with only 12% of the labels -- a setting similar to the seting in this paper --  but I don't think Lu et al. used the same 10-fold data split.  The authors of the original corpus achieved only 58% F-measure using all labels, but F-measure is not the same as accuracy.

In summary, I'm ambivalent.  This paper has no new theory, and it's not clear if the  experimental results should be considered to be good or bad, because there is no review of the literature surrounding this corpus.

---

### Official Review · AnonReviewer2 · 2020-06-29
**Interesting main idea, not good enough experimental design and description in the paper.**

**Confidence:** 5
**Rating:** 4

**Review:**

The paper presents an adaptation of an existing algorithm for semi-supervised learning, adapted for the task of audio tagging. Although the idea makes sense, the paper has flaws that are making it not good enough, according to my opinion. Please find below my major comments.

1. The paper presents a fine tuning of the hyper-parameters for an existing semi-supervised training scheme. I would be really happy to see some more work, apart from applying an existing algorithm as-is on a different modality. The paper uses the exact same formulation as in the paper of the DCT algorithm, and presents an application of it to a different dataset than the original proposed one.

2. The paper could have presented clearer the crucial details of the experimental process. Specifically, the paper mentions that there are two subsets of examples, the labeled one (denoted as S) and the unlabeled one (denoted as U). Then, the paper explicitly states that it uses examples from the union of these two subsets. This is quite confusing, since if an example has a label (thus, belonging to the S subset), then how can it belong to the U subset? If it belongs to the U, then it does not have a label and, a priori, cannot belong to the S subset.

  Similar problems exist in the evaluation setup. The paper fails to explain how the sampling of the $S\cup U$ implemented, and, additionally, it confuses the whole situation more by explicitly stating that the dataset was split in to sets, the U and S.

  Also, the paper uses quite many hyper-parameters, which it seems that are an essential component for the present method to function properly. Though, the paper does not mention explicitly how these hyper-parameters were fine-tuned, e.g. with what data, how the S and U thing was handled.

3. The paper presents results on only one model. The performance of the model is way below the state-of-the-art (SOTA) for the Urban8k. Specifically, the employed model achieves an average accuracy below 50%, where recent models go above 70%. Since the method is model agnostic, the paper should have employed a good performing model, if the SOTA was not in consideration (but the paper should have stated why it would be not in consideration).

  Having this under-performing model, cannot allow a proper evaluation of the benefit of the method. It might be that using a model that can perform well on the utilized dataset, the presented method did not offered any benefit. It might be the exactly opposite. But, nobody can say this now, only speculate.

4. Finally, the presentation of the method is not clear.

For the above reasons, I propose the rejection of the paper.

Please find below my detailed comments.

=========================================================

Line 74, right column: “mutually-exclusive classes.” The full stop before the equation, breaks the flow. Is equation 2 part of the text? There is no comma after it and the full stop before makes the reading difficult.

line 096, right column: Why $S\cup U$ is not the empty set? If it is not the empty set, then it is indicated that the examples in the $S\cup U$ are not unlabeled. So, the definition kinda breaks, because an example can be either labeled (and thus be in the S set) or unlabeled (and thus be in the U set). And, since in line 103, right column, x’ is sampled from the U set, the paper should clarify this. Especially since in line 55, right column, it is clearly stated that S and U are the subsets of labeled and unlabeled data. If an example has a label (i.e. is in the S set) then it can’t be in the U set (i.e. be unlabeled) since it already has a label.

Line 104, right column. How can a model be functioning as a typical classification model (based a typical CE loss), and (and the same time) as a generative one, that is able to generate an example as the ones getting at its input? It is not straightforward how the papers achieves that, and it should be clarified.

Line 110, left column: The paper should elaborate more on this. As it is written at the moment, it seems that the paper does proper generation of examples, and not just adding noise (even the noise according to the FGSM). It is confusing.

Line 142, left column. According to the definitions in section 2, the S and U are overlapping. Else, the method will not be able to sample the $x_{1}$ for Eq. 4. But, as it is written here, i.e. “As DCT is a semi-supervised learning method, we artificially split the training subsets into two parts: one labeled part denoted by S (for supervised) and one unlabeled part denoted by U (for unsupervised)” this is not apparent. It is not clear how the overlapping happened.

Line 115, right column: It is not clear what the “LX” is.

Line 130, right column. The paper should explain how the hyper-parameters for Eq 1 are fine-tuned. What split of data is used? What was the overlapping between S and U?

Line 134, right column. $\epsilon$ has not appeared in any equation. At what $\epsilon$ does the paper refer to?

Line 138, right column. Was Nesterov momentum used in SGD?

Line 140, right column. How the hyper-parameters for the learning rate annealing were defined? According to what data? What was the overlapping between S and U?

Table 1. It is not clear what system is used as the “Supervised”. Also, it is not clear what is the “Augmeneted”, since nothing about it was mentioned either in Section 2 or 3. Finally, the current SOTA results on Urban8K are quite (but really quite) higher than a mere 47.3. Why is the paper presenting such really really low results?

If by “supervised” is the resulting mean (I supposed, since it is not mentioned) accuracy with the same model, but with a typical supervised training, then it is evident that the model is not good for this task, since there are models (also freely available in GitHub) that perform really better. This does not allow a proper evaluation of the proposed method, since the opposing paradigm (i.e. the “supervised”) is not good, to begin with. There are pretty good chances that if the supervised method was better, then the proposed approach would not be able to surpass it. There are also chances for the opposite. But, as the set-up is at the moment, no proper evaluation is allowed.

Line 159, right column. How these percentages about the amount to the examples from S, have been decided? Based on what data?

Line 172, left column: “it falls short of what can be observed when applied to the visual object recognition task” It is not clear what is meant by this sentence.

Line 169, right column. The paper should clarify what is mean by the “noise addition” “signal processing algorithm” in “different signal processing algorithms, such as pitch shift or noise addition”. Which algorithm is meant for the noise addition?

Line 182, right column. The different hyper-parameters for the data augmentation techniques, are not clear how are fine-tuned. Also, it is not clear which of them were obtained according to previous published work. The paper should clarify, both which values of the utilized hyper-parameters were adopted according to previous, and how the other values were fine-tuned.

---

### Decision · Program_Chairs · 2020-07-01

**Decision:**

Reject

**Comment:**

Dear author(s),

Thank you very much for your submission at the ICML2020@SaS workshop (https://icml-sas.gitlab.io/). Based on the scores assigned by the reviewers, we regret to inform you that the paper was rejected. We got 26 submissions and we were only able to accept 13 papers. We invite you anyway to consider the feedback of the reviewers and to follow our upcoming workshop on July 17.